# Much Ado about Sleep: Current Concepts on Mechanisms and Predisposition to Pediatric Obstructive Sleep Apnea

**DOI:** 10.3390/children8111032

**Published:** 2021-11-11

**Authors:** Ashley L. Saint-Fleur, Alexa Christophides, Prabhavathi Gummalla, Catherine Kier

**Affiliations:** 1Department of Pediatrics, Stony Brook University Medical Center, Stony Brook, NY 11794, USA; ashley.saint-fleur@stonybrookmedicine.edu; 2Department of Anesthesiology, Stony Brook University Medical Center, Stony Brook, NY 11794, USA; alexa.christophides@stonybrookmedicine.edu; 3Department of Pediatric Sleep Medicine, Valley Hospital, Ridgewood, NJ 07450, USA; prabhavathi.gummalla@gmail.com

**Keywords:** pediatric obstructive sleep apnea, mechanisms, upper airway, critical pressure, starling resistor model, craniofacial, adenotonsillar hypertrophy, neuromuscular control, inflammatory cytokines

## Abstract

Obstructive Sleep Apnea (OSA) is a form of sleep-disordered breathing characterized by upper airway collapse during sleep resulting in recurring arousals and desaturations. However, many aspects of this syndrome in children remain unclear. Understanding underlying pathogenic mechanisms of OSA is critical for the development of therapeutic strategies. In this article, we review current concepts surrounding the mechanism, pathogenesis, and predisposing factors of pediatric OSA. Specifically, we discuss the biomechanical properties of the upper airway that contribute to its primary role in OSA pathogenesis and examine the anatomical and neuromuscular factors that predispose to upper airway narrowing and collapsibility.

## 1. Introduction

Obstructive sleep apnea (OSA) is a form of sleep-disordered breathing consisting of upper airway collapse during sleep that results in repetitive arousals and desaturations [1,2,3]. Estimates of disease prevalence range between 1% and 3% of the general pediatric population, with significant associated clinical consequences on individual morbidity [1,4,5]. Consequences of untreated pediatric OSA include metabolic, endocrine, cardiovascular, and neurobehavioral implications with potential for long-term consequences [6,7]. However, many pathogenic features of this syndrome in children remain unclear [8]. Understanding principal mechanisms of OSA is critical for ultimately allowing for the development of therapeutic strategies [2]. Much of the pathogenic mechanisms of OSA currently center around upper airway collapsibility.

The upper airway is a collapsible tube bordered by muscles controlled by reflexes that develop in early life [9]. These muscles attach to bones that encompass the orofacial region [10]. Reflex-loops activate these muscles, but during certain periods of sleep, many of these reflexes are diminished or even non-functional. This increases the propensity for airway collapse during sleep when compared to wakefulness. The pathophysiological factors involved in OSA can be compartmentalized based upon factors that promote upper airway collapsibility [11]. Both intrinsic and extrinsic factors affect this risk of collapsibility. The intrinsic factors have been studied through the evaluation of critical pressure, while extrinsic factors impacting the upper airway include anatomic and neuromuscular contributions.

Understanding of the pathophysiology in pediatric OSA assists with clinical phenotyping, which consequently assists with related therapies for anatomical considerations, neuromuscular compensation, arousal threshold, and ventilatory control [12,13]. In this article, we review current concepts surrounding the mechanism, pathogenesis, and predisposing factors of pediatric OSA. Specifically, we consider factors that contribute to upper airway narrowing and collapsibility, which affect the fine balance of airway patency during sleep (Figure 1).

## 2. Intrinsic Determinants of Upper Airway Collapsibility

Disturbances in critical pressure play a primary role in OSA pathogenesis [2]. The critical pressure elucidates the relationship between closure of the upper airway and atmospheric pressure; these upper airway biomechanics are considered intrinsic factors that contribute to upper airway patency [1].

### 2.1. Critical Pressure

Patients with OSA have been found to have increased pharyngeal critical pressure, resulting in upper airway collapsibility [14,15]. It is broadly recognized that pharyngeal collapse involves a complex interplay of neuromuscular and mechanical factors [16]. The biomechanics of pharyngeal airflow obstruction is distinguished by upper airway occlusion and collapse during sleep followed by reopening upon wakefulness [2,17]. Occlusion and the stoppage of flow transpire when upstream pressure falls below critical pressure (Pcrit) [18]. Pcrit is the pressure at which airway collapse occurs [12,19]. Adult OSA patients have significantly greater Pcrit levels [20]. Similarly, children with OSA have positive Pcrit values, while children without OSA have the ability to maintain inspiratory airflow even at strikingly sub atmospheric pressures [19].

Pediatric airways have been observed to correlate to the Starling resistor model and Pcrit has been correlated with the degree of upper airway obstruction [20,21]. The notable features of the Starling resistor model include a collapsible segment dividing the fixed resistances of the rigid segments of the upstream nasal passages and downstream trachea (Figure 2) [2,12,16,22]. These rigid segments are distinguished by intraluminal pressures, or upstream and downstream pressure, as well as upstream and downstream resistance to airflow. When the downstream and upstream pressures are less than the critical pressure, the airway closes. When the upstream pressure is greater than the critical pressure and the downstream pressure is below the critical pressure, the airway experiences inspiratory flow limitation (IFL) and can undergo a rapid oscillatory pattern between an open and a closed airway [2]. When both the upstream and downstream pressures are above the critical pressure, the airway remains open and patent [2,23].

IFL during sleep is a common occurrence that results from upper airway narrowing [24]. IFL is described as the plateauing or flattening of inspiratory flow curve over time and is a non-invasive indicator of elevated upper airway resistance [25]. Flow limitation can be found when a more negative intrathoracic pressure does not result in a corresponding increase in airflow [24]. IFL is associated with a reduction in airflow, which is then followed by arousals and leads to the activation of upper airway dilator muscles for a compensatory increase in respiratory effort [24]. Persistent airflow limitation leads to sleep fragmentation [26]. Patients with IFL or upper airway resistance syndrome have been noted to present with hypotension and elevated vagal tone [27]. Nonetheless, although critical pressure and flow limitation do not provide an indication of OSA severity, these characteristics have become highly recognized as patterns for the identification and analysis of sleep-disordered breathing [14].

### 2.2. Oscillatory Patterns

In contrast to smooth oscillations in airflow, OSA features sudden variances in ventilatory compliance and upper airway patency [2]. This notably deviates from the principles of time-invariance and linearity. Under these circumstances, ventilation is not determined by ventilatory drive, rather by the degree of upper airway patency. If ventilatory supply cannot match the demand required, the ventilatory drive significantly increases [2].

There are two primary assumptions that assist with the determination of oscillatory sleep patterns. The first point explains that ventilation responses are determined by the ventilatory demand. Inherently, when an individual experiences periodic hypoventilation and their carbon dioxide concentration increases, it leads to a linear increase in ventilatory drive [28]. Secondly, the mechanical components of respiration and ventilatory control remain comparatively constant across breathing cycles [2,28]. These two principles comprise the models that predict the smooth sinusoidal oscillations as observed in Cheyne–Stokes breathing.

OSA is notably a departure from the typical ventilatory pattern of smooth oscillations in airflow. It has been shown that obstructive apneas are characterized by dynamic pharyngeal obstruction during sleep with abrupt re-opening upon arousal [29,30]. The patency of the upper airway is quickly restored when arousal occurs, or apnea terminates, as it moves from a flow-limited-state to a non-flow-limited state [2]. These repetitive demonstrations of mismatch between ventilatory demand and ventilatory supply, created by the mechanoreceptors, chemoreceptors, ventilatory drive, and transition between the flow limited state and non-flow limited state in the upper airway, are the primary mechanism behind the oscillatory pattern in the disruption of sleep.

Interventions such as surgical interventions, positional therapy, oral sleep appliances, and pharmacotherapy conjointly assist with combating the disturbances of upper airway mechanical loads or neuromuscular compensations [31]. A comprehensive approach to treatment with multidisciplinary knowledge of the multiple intrinsic features that are responsible for upper airway collapsibility is crucial when treating these airway abnormalities [32].

## 3. Extrinsic Determinants of Upper Airway Collapsibility

Investigators have shown that airway collapsibility is usually more prominent in OSA patients under passive conditions, signifying fundamental anatomic defects when compared to matched controls [2]. These OSA patients also exhibit dampened active responses to airway obstruction, indicating associated deficits in neuromuscular control [2]. These findings suggest that the previously discussed elevations in Pcrit seen in OSA patients may be due to deficiencies in both upper airway anatomic and neuromuscular control, and these disturbances play a pivotal role in OSA pathogenesis [2]. Therefore, pathophysiology of OSA in children features the elaborate interconnection between anatomical predisposition for airway collapse and neuromuscular compensation [12].

### 3.1. Anatomic Alterations

The upper airway size is related to development of its bony supports, which are primarily coordinated by genetics and later expanded upon by environmental factors [10]. A variety of anatomic factors that contribute to increases in airway collapsibility have been identified across the life course.

#### 3.1.1. Craniofacial Growth during Fetal Life 

The earliest form of the face emerges approximately four weeks into fetal development. Migration of cranial neural crest cells, which then develop into facial prominences, is a prominent step in fetal development, and the family of homeobox genes play a role in the formation of the final tissues [10]. The facial skeleton undergoes rapid development during the first trimester of gestation and continues during the second trimester with growth of the oral cavity. The movement of the fetal tongue between the six and tenth weeks of gestation allow the closing of the primitive mouth and horizontal orientation of the tongue [10]. This organization is controlled by the 39 homeobox genes. From the third to fifth months of gestation, brain-stem neuronal networks aimed at sucking–swallowing functions are created, leading to the standard development of the oral cavity during the third trimester [10].

The final three months of gestation are a preparation period for the intricate activity of the upper airway and of reflexes needed for the critical functions of sucking and swallowing [9]. Some specific dysfunctions involving sucking, mastication, swallowing, and nasal breathing seen in premature infants impact bone development of the structures supporting the upper airway, reinforcing the concept that dysfunction leads to dysmorphism [33,34]. Studies of premature infants have shown that absence of complete upper airway training has a clear impact. Early premature infants often have irregularities involving feeding, with weakness of orofacial muscles that contribute to a small upper airway and negatively alter craniofacial growth [34,35]. Early life is an extremely susceptible period, as the rate of orofacial growth is highest between birth and two years of age and remains extremely active until six years of age [9]. Any impairment in normal orofacial function can lead to immediate and lasting impacts on growth. Because this dysfunction occurs early in life at a time when orofacial growth is at its greatest, abnormal growth may be experienced for years to come. Minimal or lack of orofacial muscle training, alongside a generalized hypotonia, does not allow for normal orofacial development. Structural growth is altered by this disturbance in normal muscle function, which leads to more constricted bony supports for the muscles forming the upper airway [9].

#### 3.1.2. Craniofacial Dysmorphism

Various craniofacial features related to skeletal morphology may predispose to upper airway collapse [2,36]. Anatomic contributions include crowded oropharynx, micrognathia, macroglossia, and midface hypoplasia [19]. Particularly, the retro-palatal region is the most common site of obstruction in pediatric patients, which can be thickened by alterations of the size and position of the mandible and tongue [8]. Micrognathia increases the risk of OSA because the tongue is larger compared to the mandibular structures and tends to prolapse backwards [33]. Another factor that correlates with the degree of OSA is the degree of midface and mandibular hypoplasia.

Genetic inheritance and functional factors influence craniofacial growth [37]. Genetic risk factors have been identified in the development of OSA to variable degrees [8,38]. Genetics are critical to the development of bone structure, which in turn plays an important role in the size of the upper airway [34]. Although a specific gene location associated with increased OSA risk has not been identified, the strongest genetic predispositions have been related to facial morphotype [38]. Dolichocephaly, or narrow face, is a familial trait that has been implicated as a risk factor regardless of ethnicity. Furthermore, there is a greater risk of OSA in a family in which a member is affected [38]. 

Functional factors, such as predominant mouth breathing, lead to significant craniofacial growth abnormalities [39]. Mouth breathing is often a result of increased nasal breathing resistance secondary to enlarged nasal turbinates or adenotonsillar hypertrophy. This impacts craniofacial growth in a developing child by altering tongue position and oropharyngeal volume [37]. The tongue tends to become low and protruded in position with a contracted upper arch, altering its muscular balance with the cheek [40]. The altered functioning of muscles induced by this change renders the maxillary arch high and narrow [41]. In healthy children, craniofacial features associated with multilevel upper airway narrowing include a small posterior airway space, retruded mandible, vertically oriented occlusal and mandibular planes, and long lower face [1,40,42]. These long, retrusive facial characteristics, known as “adenoid facies” (Figure 3), observed among mouth breathers and healthy children with OSA are hypothesized to occur because of muscular forces on the developing craniofacial skeleton.

The craniofacial features of non-mouth-breathing OSA subjects vary [40]. In these cases, Class II malocclusion with crossbite and mandibular retraction are among the most commonly reported findings [35,40,42]. Tongue posture is not associated with a contracted upper arch, as in the mouth-breathing group. Instead, arch morphology presents a wider maxillary cross-section [40]. The clinical evaluation reveals the presence of a deep upper maxilla with small and retracted jaw. For example, similar dental arch morphology during the primary dentition period was found to be a predisposing factor for OSA among children of the preschool age [41]. In their study of 16 preschool children with confirmed polysomnographic diagnosis of mild OSA, Lee et al. (2020) found that these children were more likely to present with Class II malocclusion characterized by retrognathic mandible [41]. The differential growth of the mandible relative to the maxilla was found to be an important factor during the transition from primary dentition to permanent dentition. 

Children with OSA have also been characterized as having abnormal growth hormone secretion resulting in deficient ramus growth and oral-facial hypotonia which leads to secondary changes in maxillary-mandibular growth [41]. Again, changes in maxillary-mandibular growth contributes to formation of the retrognathic mandible, which can result in reduced space between the cervical column and mandible, leading to the posteriorly postured tongue and soft palate, as well as increased risk for impaired respiratory function [41] (Figure 4).

Furthermore, tooth agenesis, whether seen in conjunction with congenital disorders or in children with early extraction of permanent teeth, has been associated with the presence of OSA [10,34]. Absent or early extraction of permanent teeth in childhood during their growth period, can result in bone retraction and affect facial bone growth. The manipulation of elements involved in orofacial growth during childhood contributes to subtle changes in bone support of the upper airway and increases the risk of collapsibility during sleep. This abnormal oral-facial growth leads to a reduction in the ideal size of the upper airway and sleep-disordered breathing with progressive worsening toward full-blown OSA [34].

In addition, children with cleft lip or palate are at an increased risk of OSA secondary to changes in the upper airway structure and nasal deformities [1,33]. Similar to healthy children with adenotonsillar hypertrophy, children with cleft lip/palate may demonstrate features of maxillomandibular retrusion and vertically oriented occlusal and mandibular planes, which compromise the airway. Children with isolated cleft palate may have significant OSA, but the severity of OSA is much greater with palatal clefts associated with craniosynostosis syndromes [33].

#### 3.1.3. Craniosynostosis Syndromes

During early fetal life, genetic mutations at the 39 homeobox genes have a major influence on the development of the upper airway region and are responsible for important malformations, such as Pierre Robin Sequence, Treacher Collins syndrome, hemifacial microsomia, and Down syndrome [9]. These syndromes are associated with abnormal breathing during sleep, as children with craniofacial anomalies related to craniosynostosis syndromes have been reported to have a high incidence of OSA [1,9]. Airway obstruction in these children can be severe and the subsequent OSA and intermittent hypoxia has serious effects on development [33]. 

There is a large amount of skeletal and soft tissue differences that distinguish each craniofacial syndrome. For example, Crouzon and Apert syndrome are characterized by the presence of a concave facial profile, while children with Pierre Robin sequence and Treacher Collins syndrome are more likely to have convex facial profiles with smaller mandibles [1]. However, unlike Crouzon syndrome, in which the midface is typically formed but retruded, the midface in Apert syndrome is underdeveloped as well as retruded [43].

Despite their differences, the craniofacial skeletal similarities, including short anterior skull base measurements, may be associated with OSA. Namely, the anterior skull base forms the roof of the orbit and midface and is responsible for the depth and projection of the midface [1]. Furthermore, children with Apert syndrome, Treacher Collins syndrome, and Pierre Robin sequence are more likely to have a steeply inclined mandibular plane and retrognathia, which can be associated with a small posterior airspace [1]. Children with Apert syndrome also often have dental anomalies, such as tooth agenesis of the maxillary canines, high arched palate, cleft palate, and difficulty with nasal breathing, which all result in predisposition to OSA [43].

Similarly, children with Down Syndrome are more likely to have OSA secondary to their craniofacial anatomy and generalized hypotonia [7,33]. Typically, these children have midface hypoplasia, glossoptosis, macroglossia, small airways, reduced upper airway tone, and obesity [33,44]. These factors increase their risk of inspiratory obstruction in the upper airway during sleep [33]. In their examination of 188 children with Down Syndrome, Hill and colleagues found that 14% had moderate to severe OSA and 59% had mild OSA confirmed on polysomnography, which have many implications for long-term morbidity and mortality [44]. 

#### 3.1.4. Pharyngeal Soft Tissue

The classic risk factor for OSA is an enlargement of the adenoids and tonsils [19,36,40,42,45,46,47]. In fact, adenotonsillar hypertrophy peaks between three to seven years of age, coinciding with the peak incidence of childhood OSA [1,19]. Compared to controls, children with OSA have larger adenoids, tonsils, and soft palates. Soft tissue hypertrophy is a common cause of upper airway narrowing in children with OSA, evaluated by the Friedman Tonsillar Grading Scale [48]. Children with Size 4 tonsils (kissing tonsils) are extremely susceptible to the development of sleep disorders (Figure 5). Adenotonsillar hypertrophy is a contributor to narrowing of the retro-palatal area, which has the smallest cross-sectional area, making it the most frequent site of obstruction.

A crucial factor in the genesis of OSA in these children is the increase in soft tissue volume [49]. Fat deposits that infiltrate the tongue muscles, neck muscles, and surrounding soft tissues clearly reduce the upper airway lumen [9]. During the rapid growth period of childhood, reducing the size of the upper airway leads to disturbance of sleep and increasing upper airway resistance. Mouth breathing occurs and worsens the entire situation by further contributing to this increase in resistance and abnormal orofacial growth [50]. Habitual snoring develops and begins the slow progression towards OSA [9].

Recent studies have demonstrated that the cephalometric stereotype associated with respiratory dysfunction vary in relation to the obstructive tissue involved. The long face stereotype mainly corresponds to the adenoidal hypertrophy, while tonsillar hypertrophy is mainly characterized by a tendency towards a more horizontal mandibular growth, counterclockwise rotation of the jaw, and a higher ratio between the posterior and anterior facial heights [40]. When adenoid tissue hypertrophy obstructs airflow, children tend to extend the head to increase retroglossal and hypopharyngeal airway volume, which leads to a greater vertical growth pattern. In addition, these children also resort to mouth breathing, which results in posterior tongue movement and buccal musculature shifting medially against the maxilla, contributing to maxillary constriction [1].

A dynamic inspiratory airway narrowing during tidal breathing has additionally been noted to affect the pharyngeal airway volume in children with OSA [36]. These findings stipulate that patients with OSA must either have local narrowing of the airway or increased airway compliance. Nasal resistance, a good indication of local narrowing, has been reported to be increased in children with OSA compared with controls [36]. Arens et al. (2011) found that even in asymptomatic children, greater adenotonsillar size correlated with increased nasopharyngeal airway narrowing during inspiration [51]. The narrowest airway segment occurs at the site of the overlap between the tonsil and adenoid. However, the collapse can happen at different levels of the pharynx [36]. Isono et al., (1998) examined different sites of collapse during paralysis in both children with OSA and controls, demonstrating that negative pressure is required to collapse the airway in controls, but this is not necessary in children with OSA [52]. Furthermore, children with OSA had not only nasopharyngeal collapses but also a generalized increased collapsibility of the pharynx when compared with normal controls. These anatomic variants are believed to increase Pcrit by restricting the size of the bony enclosure around the pharynx and/or increasing the volume of enclosed soft tissue, which all result in increased upper airway collapsibility [2,12,53,54,55,56].

#### 3.1.5. Puberty-Related Changes

Puberty is another period of high orofacial growth velocity [9]. Growth and additional changes during puberty have an important influence on OSA progression and development as the human face undergoes a 2.5-fold increase in size from birth to adulthood [36]. At puberty, growth involves a counterclockwise rotation of the mandible with vertical increase of the maxilla. Corrective action targeting abnormal orofacial growth should occur prior to onset of these pubertal changes to prevent further negative consequences on the orofacial development. For example, abnormalities in the manner in which the upper teeth meet the lower teeth can lead to an overbite. If this has not been adequately treated, the upper teeth later block the counterclockwise rotation of the mandible, creating a retro-position of the mandible and additional narrowing of the upper airway [9]. Furthermore, with puberty and increased hormone secretion, an enlargement of muscles and soft tissues occurs, particularly among boys. These structural changes contribute to the progressive development of habitual snoring with the associated negative impact on upper airway [9].

Ronen et al. (2007) investigated the impact of age and sex on upper airway length in children and adolescents [57]. This is of special interest because longer airways are more susceptible to OSA [57]. This study offered insight into possible mechanisms for the observed male predominance seen in adult OSA, but not in pediatric OSA. The authors found that prepubertal boys and girls have similar upper airway lengths and similar OSA prevalence. However, upper airway length increases significantly in boys during puberty. This increased length, along with testosterone secretion in boys, have both been noted to increase upper-airway collapsibility [57]. In contrast, estrogen and progesterone in girls have both shown to be protective against upper airway collapse, for unclear reasons. These findings have implications for the pathophysiology of OSA, partially explaining the similar gender prevalence in early childhood but strong male predominance in adulthood [57]. The patterns of disruption in normal airway growth and development could also have suggestions for future occurrence of OSA.

#### 3.1.6. Obesity

Obesity is also a significant anatomical risk factor for upper airway obstruction during sleep [2,3,46,51,58]. Obesity is defined as BMI above the 95th percentile and affects OSA mainly through two mechanisms: (1) fat at the pharyngeal soft tissue reduces the upper airway lumen, increasing structural collapse, and (2) increased abdominal adiposity significantly reduces respiratory function [8]. The upper airways of obese individuals are more prone to collapse as there are proportional elevations in Pcrit with increasing BMI [2]. In their case-control study comparing anatomical findings in obese children with OSA to those without OSA, Arens et al. (2018) found larger parapharyngeal fat in obese children with OSA, but there was not a direct association with OSA or obesity severity [51].

In addition, obese individuals have smaller lung volumes, which leads to decreased amount of caudal traction on the upper airway and an increased critical closing pressure. This is particularly prevalent in patients with abdominal adiposity, which has been shown to decrease lung volume nearly to the level of residual volume [2]. Conversely, weight loss has been shown to produce improvements in OSA outcomes through a reduction in surrounding tissue pressure and an increase in caudal traction, both of which decrease Pcrit [2]. Although adenotonsillenctomy remains the first-line therapy for pediatric OSA, including in obese children, weight loss as an adjunctive therapy is critical in the treatment of obesity-related OSA, particularly among those who have underwent adenotonsillectomy without significant clinical improvement [8].

### 3.2. Disturbances in Neuromuscular Control and Inflammatory States

Although anatomic defects are known to contribute to the pathogenesis of OSA, these structural alterations may only account for one-third of the variability in OSA severity [2]. Evidence for the involvement of other processes besides anatomic defects is found in work by Guilleminault et al. (1989), as they described initial resolution of OSA following tonsillectomy in a cohort of children, but recurrence in adolescence [59]. Other studies have demonstrated that some children with adenotonsillar hypertrophy do not develop OSA [19]. These findings suggest that there are additional factors that play a role in a patient’s predisposition to developing OSA. While anatomic obstruction undoubtedly contributes significantly in some children, neuromotor factors that affect ability to maintain airway patency and ventilation during sleep may predominate in others [1,19]. In fact, a failure of the upper airway muscles to generate an appropriate response to airway collapse could be caused by an afferent inability to sense the airway narrowing, an efferent motor control problem, or both. Furthermore, inflammatory states, both local and diffuse, may be related to the mouth breathing that predisposes to OSA.

#### 3.2.1. Ineffective Upper-Airway Dilator Muscles

Reductions in neuromuscular tone are suspected to contribute to increased OSA severity [2]. During wakefulness, reflex activation of the upper airway dilator muscles prevents collapse and maintains upper airway patency [60]. However, these protective mechanisms are deranged during sleep. Upper airway muscle collapse is highly dependent on the dynamic balance between intraluminal pressures and neural drive [23,61]. This vulnerability is compounded in individuals who also have an anatomic susceptibility to airway collapse.

There are complex patterns of neural activation in the upper airway that differ between muscles. For example, the genioglossus muscle, located at the base of the tongue, is the largest pharyngeal dilator muscle. It receives several sources of input, including centrally from the brainstem and reflexively from pressure sensitive mechanoreceptors in the upper airway and chemical drive via hypoxia [61,62]. Its activity varies between sleep stages. This is in contrast to the tensor palatini muscle, a palatal muscle, which is less sensitive to changes in pharyngeal pressure and across sleep stages when compared to the genioglossus muscle. However, it can be activated to a similar degree with large pressure swings. The two-fold effect of a loss of central drive along with reflex input to the upper airway muscles during sleep are thought to be important drivers in OSA pathogenesis. A significant proportion of OSA patients do not increase genioglossus activity or only increase by a small amount when exposed to airway narrowing [62]. 

The inability to generate an appropriate response can also be related to local afferent control. A gradual neuropathy of the soft palate and pharyngeal dilators could be associated with the progression of habitual snoring to OSA [63]. Neuronal degeneration due to habitual snoring can lead to relaxation of pharyngeal dilator function during sleep. The relaxation of these muscles is a factor of recurrent obstructions and fragmented sleep. The neuronal degeneration and loss of afferent neuronal activity in the upper airway can occur as a result of long-term vibratory trauma, which is exemplified during habitual snoring [63]. Being that snoring is produced by turbulent flow of air vibrating the soft palate, it is plausible that the vibratory trauma and its associated inflammatory changes could culminate in neuronal damage [63]. Furthermore, sensory testing studies demonstrate progressive neuronal dysfunction along the spectrum of habitual snoring and OSA, which also contributes to afferent dysfunction of the palate [63,64].

#### 3.2.2. Reduced Tongue Mobility

Accumulating evidence suggests that tongue mobility plays an important role in maintaining airway patency [65]. When there is limited mobility and insufficient upward force exerted by the tongue to the upper palate, the individual is more prone to developing the abnormally narrow V-shaped maxilla, unlike the normal U-shaped maxilla. This leads to a narrow nasal floor and increases the risk of nasal obstruction [65]. Tongue mobility also correlates with the relative position of the mandible, which may be an important factor influencing mandibular development [65]. Tongue motor immaturity is defined as the inability to elevate the tongue against the palate and is associated with increased symptoms of sleep-disordered breathing after adenotonsillectomy [65].

Furthermore, the tongue has many receptors on its surface that allow for the perception of touch, movement, and position. Sensory feedback is required for motor skill, including contraction of the buccal muscles and appropriate tongue movement [66]. Abnormal stimulation or lack of stimulation of orofacial structures contributes to speech apraxia, as the normal development of the motor plans for speech is concurrent with normal stimulation of orofacial growth [66].

Recently, orofacial myofunctional therapy (OMT) has been suggested as an adjunct treatment of OSA in children [67]. OMT is aimed at targeting abnormal breathing patterns and muscular dysfunction that can impact upper airway patency [68]. It also aims to enhance the tongue position and strength [65]. This is achieved through oropharyngeal exercises that focus on the dysfunctional upper airway muscles. For example, therapies aimed at appropriate tongue positioning include forced pressing of the tongue against the palate or placing the tip of the tongue against the palate and sliding the tongue backward [69].

#### 3.2.3. Unstable Ventilatory Control and Low Respiratory Arousal Threshold

Unstable ventilatory control may contribute to OSA via several mechanisms. Loop gain is a term often applied to engineering, describing the sensitivity of a response to a given stimulus. When extrapolated to respiratory pathophysiology and obstructive sleep apnea, it describes the ratio of the ventilatory response to the disturbance that elicited the response [19]. Patients with high loop gain are more prone to the oscillations in breathing seen in sleep apnea because they have overly sensitive, unstable ventilatory systems with poor upper airway muscle recruitment [62]. For example, small or intermittent hypoxic events (less than 2 mmHg change in carbon dioxide) produces a large ventilatory response in these patients. In fact, a large negative inspiratory pressure ensues, which closes the pharyngeal airway. In addition, oscillations in respiratory drive due to high loop gain can produce ensuing periods of low respiratory drive [62]. Whereas an increase in ventilatory drive promotes airway patency by activating and recruiting upper airway muscles, low respiratory drive has an opposite effect [19,62]. As such, patients with unstable ventilatory control may experience fluctuations in ventilatory drive, disrupting the upper airway and promoting collapse when ventilatory drive is at its lowest [19].

The arousal threshold is an important concept defining the propensity of an individual to wake up from sleep [70]. Low respiratory arousal threshold, which has been implicated in OSA pathogenesis, is defined as the occurrence of arousal from sleep with a small rise in ventilatory drive, meaning an individual more easily awakens [70,71]. In contrast, an individual with a predisposition to OSA secondary to anatomic considerations may be protected by a high respiratory arousal threshold that allows for accumulation of adequate respiratory stimuli to promote stable breathing [70]. On the other hand, an individual with similar anatomy and a low respiratory arousal threshold would present with repetitive apnea. Although it was once thought that arousals were necessary to restore airflow following an apneic episode in OSA, it is now clear that arousals likely propagate the cyclical breathing patterns that occur in OSA [62]. The reasons why readily awakening due to airway narrowing may contribute to OSA pathogenesis are three-fold. First, frequent arousals can prevent deeper, more stable stages of sleep. Secondly, premature arousal limits the adequate buildup of respiratory stimuli needed to activate the upper airway muscles [71]. Lastly, arousals from sleep can produce a sudden rise in minute ventilation that drives carbon dioxide levels below the apnea threshold and perpetuate unstable ventilatory control [62]. However, compared with adults, children have higher respiratory arousal threshold, so there is evolving evidence that arousals may be insufficient markers of sleep disruption in children [72,73].

#### 3.2.4. Local Inflammation and Allergic Rhinitis

Mucosal inflammation may dull local sensory input and neuromuscular responses to upper airway obstruction, leading to more prominent upper airway obstruction during sleep [2]. Allergic rhinitis in particular can affect sleep through several mechanisms. Nasal congestion secondary to the nasal mucosa inflammation, mucosal edema, and secretions can induce increased airway resistance, contributing to mouth breathing, sleep disruption, and fatigue [8,40,46,74,75].

#### 3.2.5. Diffuse Inflammation

Histological studies have demonstrated diffuse inflammatory changes in OSA patients [19,64,76,77,78,79]. The concept combining inflammation and pediatric OSA was first invoked primarily by Tauman et al., in 2004 [80]. Since then, a growing body of evidence has amassed and confirmed this association, particularly among obese children, as obesity serves as a chronic low-grade inflammatory disorder [81,82,83,84]. However, even in non-obese children, cytokines are thought to play a role in augmenting or triggering inflammation [85].

Specifically, the expression of inflammatory cytokines, such as interleukin (IL)-17, IL-23, C-reactive protein (CRP), tumor necrosis factor alpha (TNF-a), IL-1beta, IL-6, and IL-10, are heightened in children with OSA [8,19,86,87,88]. The inflammatory role of IL-17 and IL-23 has been particularly emphasized [85]. These activated inflammatory cells produce various cytokines, leading to inflammatory responses such as neutrophil recruitment, tissue destruction, and neovascularization. IL-17 also takes part in neutrophilic inflammation of the pulmonary system and leads to chronic airway inflammation [77].

Furthermore, several studies have also indicated that the NF-kB related inflammatory pathway is associated with the pathogenesis of OSA. NF-kB is an essential mediator of inflammation and the upregulated expression observed in cases of OSA increases downstream inflammatory mediators and cytokines, such as TNF-a, IL-6, and CRP. Furthermore, anoxia produces oxygen radicals and activates NF-kB, allowing it to translocate into the cell nucleus to enhance damaging transcription [89]. The investigation conducted by Lu et al. (2017) indicated that NF-kB protein expression was increased in patients with OSA and was positively correlated with the severity of disease, suggesting that NF-kB contributes to the initiation of inflammation [89].

#### 3.2.6. Sickle Cell Disease 

Sickle cell disease (SCD) is another inflammatory state that has implications for the development of pediatric OSA. In fact, there is a higher prevalence of obstructive sleep apnea in children with sickle cell disease [90]. OSA and SCD share common pathophysiological mechanisms, and hypoxia is the central trigger. In SCD, hypoxia and resultant tissue ischemia is exacerbated by obstructive sleep apnea, which sets up a vicious cycle of oxygen deprivation and inflammation [91]. Progressive damage and splenic sequestration seen in SCD leads to auto-splenectomy, often by the time these children are five years of age. This auto-splenectomy leads to a compensatory hypertrophy of secondary lymphoid tissue, which includes adenotonsillar hypertrophy. As such, treatment of OSA with adenotonsillectomy in children with SCD is an effective option for decreasing complications of SCD [91].

## 4. Conclusions

The mechanism and pathogenesis of pediatric obstructive sleep apnea is greatly influenced by the factors that lead to the increased propensity for upper airway collapsibility during sleep. This review demonstrated that abnormal changes in anatomical supports and neuromuscular changes increases the risk of collapse, which in turn leads to OSA in children. The intrinsic and extrinsic factors that predispose to OSA begin during the fetal period and continue throughout the life span. History and examination of patients, especially early on in childhood, assist with diagnosis and management of the condition [92,93]. Recognition of these underlying factors may lead to early recognition and pave the way for targeted therapies early in life aimed at decreasing the prevalence of pediatric OSA.

## Figures and Tables

**Figure 1 children-08-01032-f001:**
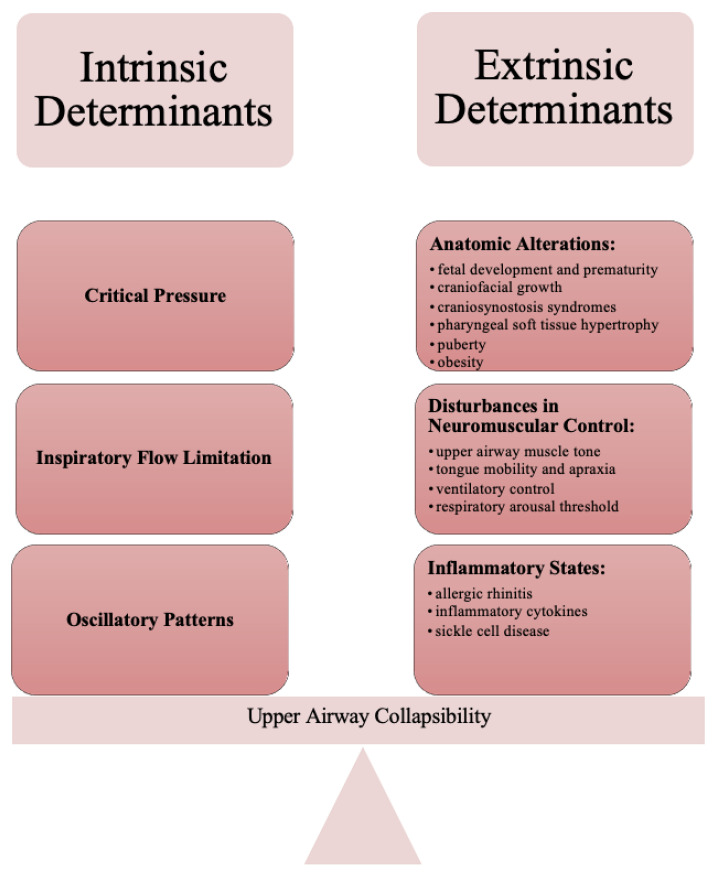
Intrinsic and Extrinsic Determinants of Upper Airway Collapsibility that affect the fine balance of airway patency during sleep.

**Figure 2 children-08-01032-f002:**
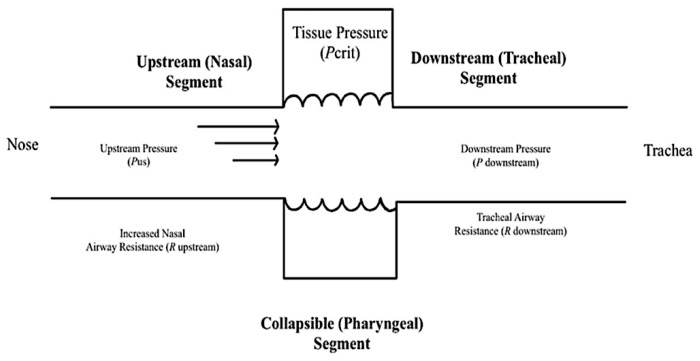
The Starling Resistor Model is a diagram that assists with the explanation of upper airway dynamics and the management of obstructive sleep apnea. The oropharynx is presented as a collapsible tube. Increased nasal airway resistance increases the probability that pharyngeal pressure will decrease below the CRITICAL pressure (Pcrit) and close the pharynx, which is a large proponent of obstructive events [2].

**Figure 3 children-08-01032-f003:**
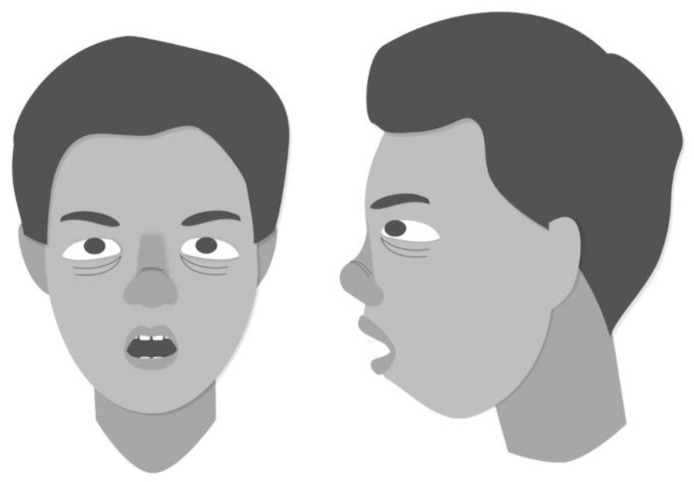
This child has “adenoid facies”, a term commonly used to describe facial characteristics of a mouth-breathing child with anterior face, retrusive mandible, sunken eyes, naroow pinched nostrils, and open mouth with crowded teeth [1].

**Figure 4 children-08-01032-f004:**
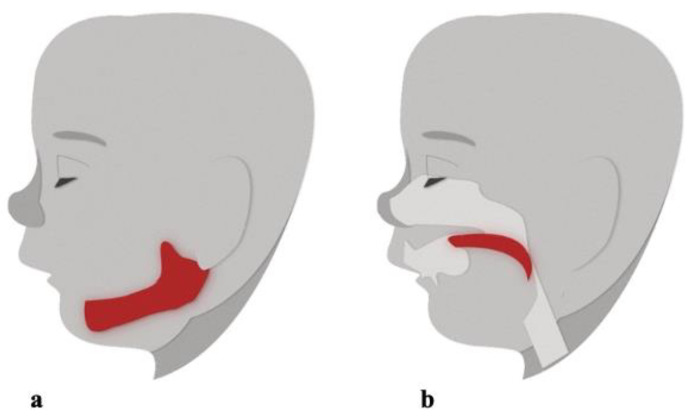
Changes in maxillary-mandibular growth contributes to formation of the (**a**) retrognathic mandible, which may result in decreased space between the cervical column and mandible, leading to the (**b**) posteriorly postured tongue and soft palate, resulting in airway obstruction [41].

**Figure 5 children-08-01032-f005:**
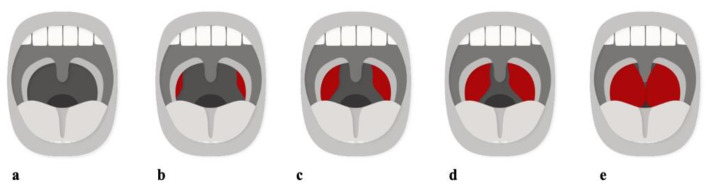
Tonsil Grading System. (**a**) Size 0, absence of tonsillar tissue. (**b**) Size 1, tonsils within the pillars. (**c**) Size 2, tonsils extended to the pillars. (**d**) Size 3, tonsils extended past the pillars. (**e**) Size 4, tonsils extended to the midline. Adapted from Friedman [48].

## Data Availability

Not applicable.

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
