# Peer review of "Much Ado about Sleep: Current Concepts on Mechanisms and Predisposition to Pediatric Obstructive Sleep Apnea"

_children, 2021, doi:10.3390/children8111032_

Round 1

Reviewer 1 Report

This is a review article on the pathogenesis and risk factors of pediatric OSA. I want to congratulate the authors on their efforts. The review article is organized and discusses several essential themes in pediatric OSA. 

Comment 1: Although Figure 1 has references in the text, I believe it also merits a citation in the figure (for example, Figure 2) unless the authors developed this figure. 

Comment 2: Line 77 Inspiratory flow limitation is used. However, its abbreviation is not used until like 80. The authors may consider editing.

Comment 3: Consider changing snoring to a more specific term such as habitual snoring in lines 295 and 413.

Comment 4: The authors should consider sickle cell anemia in the examples of inflammation in OSA development as it has both pathogenic and therapeutic implications. 

Comment 4: Since the review article is dense, the authors may consider creating a summary figure highlighting the key points of the article.  

Author Response

Thank you for the opportunity to submit our revised Manuscript ID: children-1412457 entitled, “Much Ado About Sleep: Current Concepts on Mechanisms and Predisposition to Pediatric Obstructive Sleep Apnea?” for further consideration to the special issue on Obstructive Sleep Apnea Syndrome in Children of Children. We are grateful for the many helpful suggestions made by the reviewers. We believe that in addressing these comments we have improved the manuscript. Below we describe the changes made in response as well as their location within the manuscript.

Comment 1: Although Figure 1 has references in the text, I believe it also merits a citation in the figure (for example, Figure 2) unless the authors developed this figure. 

Response: A citation has been added to Figure 1.

Comment 2: Line 77 Inspiratory flow limitation is used. However, its abbreviation is not used until like 80. The authors may consider editing.

Response: The abbreviation for inspiratory flow limitation (IFL) is used when it is first mentioned now on Line 78.

Comment 3: Consider changing snoring to a more specific term such as habitual snoring in lines 295 and 413.

Response: The more specific term ‘habitual snoring’ is used instead of ‘snoring’ throughout the text.

Comment 4: The authors should consider sickle cell anemia in the examples of inflammation in OSA development as it has both pathogenic and therapeutic implications. 

Response: A newly added Section 3.2.6, entitled Sickle Cell Disease, addresses the relationship between OSA and sickle cell disease (Lines 538-549).

Comment 5: Since the review article is dense, the authors may consider creating a summary figure highlighting the key points of the article.  

Response: The authors now include a figure highlighting the intrinsic and extrinsic determinants of upper airway collapsibility that are discussed in the text (Figure 1).

Reviewer 2 Report

This manuscript is a comprehensive review on Current Concepts on Mechanisms and 2 Predisposition to Pediatric Obstructive Sleep Apnea.

It is an interesting and well-written paper, including many updated references.

I have some minor suggestions:

- Adding a table including all the physiopathological issues discussed.

- Adding a figure illustrating them.

- Adding a parapragh on reduced tongue mobility and apraxia as a risk factor to develop childhood OSA (after lines 420…).

Yuen HM, Au CT, Chu WCW, Li AM, Chan KC. Reduced Tongue Mobility: An Unrecognised Risk Factor of Childhood Obstructive Sleep Apnoea. Sleep. 2021 Aug 25:zsab217. doi: 10.1093/sleep/zsab217.

- And its implications for myofunctional therapy

de Felício CM, da Silva Dias FV, Folha GA, de Almeida LA, de Souza JF, Anselmo-Lima WT, Trawitzki LV, Valera FC. Orofacial motor functions in pediatric obstructive sleep apnea and implications for myofunctional therapy. Int J Pediatr Otorhinolaryngol. 2016 Nov;90:5-11. doi: 10.1016/j.ijporl.2016.08.019.

Author Response

Thank you for the opportunity to submit our revised Manuscript ID: children-1412457 entitled, “Much Ado About Sleep: Current Concepts on Mechanisms and Predisposition to Pediatric Obstructive Sleep Apnea?” for further consideration to the special issue on Obstructive Sleep Apnea Syndrome in Children of Children. We are grateful for the many helpful suggestions made by the reviewers. We believe that in addressing these comments we have improved the manuscript. Below we describe the changes made in response as well as their location within the manuscript.

Adding a table including all the physiopathological issues discussed. Adding a figure illustrating them.

Response: The authors now include a figure highlighting the intrinsic and extrinsic determinants of upper airway collapsibility that are discussed in the text (Figure 1).

Adding a paragraph on reduced tongue mobility and apraxia as a risk factor to develop childhood OSA (after lines 420…).

Yuen HM, Au CT, Chu WCW, Li AM, Chan KC. Reduced Tongue Mobility: An Unrecognised Risk Factor of Childhood Obstructive Sleep Apnoea. Sleep. 2021 Aug 25:zsab217. doi: 10.1093/sleep/zsab217.

Response: Thank you for this comment. A new subsection (3.2.2. Reduced Tongue Mobility) has been added and Lines 435-450 discusses the impact of reduced tongue mobility and apraxia.

And its implications for myofunctional therapy

de Felício CM, da Silva Dias FV, Folha GA, de Almeida LA, de Souza JF, Anselmo-Lima WT, Trawitzki LV, Valera FC. Orofacial motor functions in pediatric obstructive sleep apnea and implications for myofunctional therapy. Int J Pediatr Otorhinolaryngol. 2016 Nov;90:5-11. doi: 10.1016/j.ijporl.2016.08.019.

Response: Lines 451-458 in the new subsection 3.2.2 discuss orofacial myofunctional therapy and its impact on tongue mobility.